# Intake of Radionuclides in the Trees of Fukushima Forests 2. Study of Radiocesium Flow to Poplar Seedlings as a Model Tree †

**Shoko Aoki [1], Miki Nonaka [1], Chisato Yasukawa [1], Masateru Itakura [1], Masaharu Tsubokura [2], Kei'ichi Baba [3], Hiroya Ohbayashi [4], Tomoko Seyama [4], Iwao Uehara [4], Rumi Kaida [1], Teruaki Taji [1], Yoichi Sakata [1] and Takahisa Hayashi [1,*]**

1 Department of Bioscience, Tokyo University of Agriculture, Tokyo 156-8502, Japan
2 Department of Radiation Protection, Minamisoma Municipal General Hospital, Minamisoma 975-0033, Japan
3 Research Institute for Sustainable Humanosphere, Kyoto University, Kyoto 611-0011, Japan
4 Department of Forest Science, Tokyo University of Agriculture, Tokyo 156-8502, Japan
* Correspondence: takaxg@nifty.com
† This paper is dedicated to Peter Albersheim who died on 23 July 2017.

**Abstract:** After the nuclear power plant accident in Fukushima, radionuclides were deposited over a large area of local forest. However, almost nothing is known about radionuclide infiltration into trees. Here, we used poplar seedlings as a model to show that radiocesium can enter directly into leaves and bark, moving via ray cells through the symplastic pathways to the xylem and concentrating around the meristems, cork, and vascular cambium. During induced potassium incorporation and reduced seasonal growth, the radiocesium in the meristems of stems mainly passes into abscission tissues such as leaves and heartwood. There is no turnover of radiocesium after it enters the heartwood.

**Keywords:** heartwood; perforated xylem; radiocesium infiltration; radiocesium movement; xyloglucan

## 1. Introduction

The accident at the Fukushima-Daiichi nuclear power plant in Japan in March 2011 caused the dispersal of abundant radionuclides into the atmosphere and ocean [1]. The radiocesium was markedly deposited onto trees and local residences in an aerosol form that was partly absorbed by rain or melting snow [2–4]. The radionuclide moved from the surface bark to the inner xylem in tree samples obtained from September 2011 to November 2015, 6 to 51 months after the accident [5]. Through the forest tree analyses, we found that radiocesium could be incorporated into the surfaces of trees and also could migrate throughout the tissues of the entire tree. Growing cells within the tree could be a target for the accumulation of the radionuclide, similar to that of potassium. However, its turnover in a tree body occurs via transition to the dead cells, and it also occurs in a forest field [5,6]. Nothing is known about radiocesium infiltration into trees, and very little is known about its movement in trees. This paper describes a study on the immediate effects of radiocesium on trees.

After the Hiroshima atomic bombing, radionuclides such as $^{14}C$ translocated to the tree ring of the fallout year and $^{90}Sr$ levels were insignificant from the sapwood to the heartwood [3]. However, in Fukushima forests, the xylem of two Japanese cedar trees showed radiocesium migration to the sapwood in one tree and to the heartwood in the other tree in the same field at the same time [5]. The radionuclide appears to move to the heartwood if the color of the heartwood is black, which indicates that it contains high potassium levels. It is likely that the movement is also dependent on their potassium content. This paper demonstrates model experiments on how radiocesium could move into forest trees during seasonal growth in the presence of high and low potassium levels in the soil.

## 2. Materials and Methods

### 2.1. Generation and Cultivation of Poplar Seedlings

Seedlings of wild white poplars (*Populus alba* L.) and their transgenic lines overexpressing xyloglucanase (AaXEG2, AY160774) with the signal peptide of PaPopCel1 under the control of the CaMV35S promoter were reported previously [7]. They were cultured in half Murashige and Skoog (MS) medium containing 1% agar in the absence of sucrose in a sterile plant box with an 18-h light (5000 lux)/6-h dark photoperiod at 25 °C in a growth chamber. The seedlings were propagated via root cuttings followed by transferring the shoot segments to the MS agar medium. Approximately 10-cm-tall seedlings were potted as described below. They were grown in a greenhouse under natural light conditions, supplemented with fluorescent lamps, at around 25 °C.

### 2.2. Radiocesium Incorporation

$[^{137}Cs]CsCl$ (2.5 kBq/mmol) was obtained in 2013 from Eckert & Ziegler (Valencia, USA). One microliter of 0.1 μM $[^{137}Cs]CsCl$ (2.5 MBq/mmol) in 20 mM MES/NaOH buffer (pH 6.2) was applied by placing it on part of the leaf or the stem of white poplar seedlings. For over 10 min to 24 h, the seedlings were fixed on thick paper with adhesive tape after washing the roots with water to remove the soil. Then, the fixed seedlings were microwaved at 1000 W for 10 s by NE-KM264-FG (Panasonic, Osaka, Japan) and subjected to autoradiography using an Image Analyzer (Fujifilm, Tokyo, Japan) to measure the total $^{137}Cs$ incorporation.

### 2.3. Perforated Poplar Seedlings and Cypress Stem

The lower parts of white poplar seedling stems (approximately 20 to 30 cm in height) were perforated by a drill bit, receiving seven holed incisions at less than 12 cm above the soil. A hand drill was used to make a diagonal hole approximately 1.0 mm in diameter through the stem. Then, the resulting holes were sterilized with 6.5% $H_2O_2$ [8]. The stems were then shortened to 12 cm by cutting off the upper sections. They were then cultured for one month, during which time one to two sprout shoots formed off each seedling's main stem (stump). Control untreated stem seedlings were also prepared in the same manner as described above, without any drilling.

A 15-year-old cypress (*Chamaecyparis obtuse* Endl.) tree in Minamisoma at 37°38′16″ N/140°54′13″ E [5] also received incisions on March 2013. A power drill was used to make incisions that were approximately 0.6 cm in diameter and 10 cm into the stem at less than 0.6 m above ground level. Then, the resulting holes were sterilized with 6.5% $H_2O_2$. The tree was cut down on December 2013. The trunk section that had incisions was transversely cut into 1-cm-thick sections and subjected to autoradiography with an Image Analyzer (Fujifilm, Tokyo, Japan).

### 2.4. Radiocesium Flow Experiments

After several cultivations for each of the three seedlings, all seedlings were completely fractionated into their component tissues. A 10-mm apical stem section was taken from the tops of the shoots, and the cambium was collected by scraping off both the surface of the xylem and the inside of the bark with a blade from the stem. The radioactivity of each tissue was determined using a Beckman LS Gamma Counter (Long Island Scientific, East Setauket, USA) and is shown as the relative change (%) in the total tissue content of each tree. The $^{137}Cs$-applied bark (approximately 8 mm × 8 mm) was not used for the analysis, but it was used as the standard of normalization for the radiocesium that was incorporated.

### 2.5. Microscopy for Radiocesium Autoradiography

Additionally, 10 μL of 0.1 μM $[^{137}Cs]CsCl$ (2.5 MBq/mmol) in 20 mM MES/NaOH buffer (pH 6.2) was applied for 30 min to the surface of the stems at approximately 5 cm above the soil, the poplar stems were washed with water, and stem sections (5 mm in length) were excised around the application

zone. The 5-mm sections were dipped twice in *n*-butanol for 3 min each and embedded in paraffin at 70 °C. Paraffin-embedded stems were transversely sliced by hand. The cut sections were placed onto glass slides and washed with xylene to remove the paraffin. Then, the sections were mounted on glass slides and covered with Ilford K5D emulsion film (Ilford Photo, Cheshire, UK) [9]. The glass slides were stored at 4 °C for two months for the poplar stem sections. The films were developed using Kodak D-19 (Kodak, Rochester, USA) for approximately 5 min, rinsed with water, fixed with Kodak fixer at 20 °C, washed in running water (15 min), and sealed with 50% glycerol. Slides were examined under bright-field illumination and photographed. Control poplar stems were prepared without the deposition of [$^{137}$Cs]CsCl. The glass slides were stored at 4 °C for two months.

## 2.6. Cell Wall Analysis

For chemical analysis, each tissue fractionated from poplar seedlings was homogenized in liquid nitrogen, and the resulting powder was ground in 20 mM sodium phosphate buffer (pH 6.2) using a mortar. The cell wall residue was washed three times. Each wall residue sample was washed with water once and extracted four times with 24% KOH containing 0.1% NaBH$_4$. The alkali-soluble fraction was neutralized, dialyzed with Spectra/Por dialysis membrane MWCO: 1000 (Spectrum, Rancho Dominguez, USA), and freeze-dried for use in methylation analysis [10]. Partially methylated alditol acetates were analyzed using an Agilent gas chromatography-mass spectrometer (Agilent, Santa Clara, USA) with a glass capillary column DB-225 (0.25 mm i.d. × 15 m). Each alditol acetate was identified by its retention time and mass spectrum [11]. Xyloglucan content was calculated as the 2.86-fold amount of 4,6-linked glucose in the alkali-soluble fraction because of the xyloglucan structure in white poplar [12]. The alkali-insoluble wall residue (cellulose fraction) was washed twice with water and extracted with acetic/nitric reagent (80% acetic acid/concentrated nitric acid, 10:1) in a boiling water bath for 30 min [13]. The resulting insoluble material was washed in water, freeze-dried and weighed to determine the amount of cellulose. Lignin content was determined using the Klason method [14]. The potassium content was determined using an Atomic AA-7000 Absorption Spectrophotometer (Shimazu, Kyoto, Japan) at a wavelength of 766.5 nm after digestion with HNO$_3$ and H$_2$O$_2$.

## 2.7. Immunostaining

For immunofluorescence labeling, stem samples were fixed in 3% glutaraldehyde in 70 mM sodium phosphate buffer (pH 7.0) overnight at 4 °C and then sliced into 200 μm-thick longitudinal sections using a freezing/sliding microtome at −20 °C. The sections were incubated with 50% sodium hypochlorite for 10 min to remove the protoplasm. After five 10-min washes with sodium phosphate-buffered saline (PBS), the sections were incubated in a blocking solution of PBS containing 1% bovine serum albumin for 1 h. They were then incubated in a 20-fold dilution of anti-xyloglucan antibody CCRC-M1 (CarboSource, Athens, USA) for 2 h and washed with PBS containing 0.5% *v/v* Tween-20 [15]. For the indirect immunofluorescence detection of xyloglucan, the sections were incubated in a 50-fold dilution of anti-mouse IgG antibody conjugated with Alexa 488 (Molecular Probes, Eugene, USA) for 2 h. The sections were washed and subsequently embedded in an antifading reagent (Molecular Probes, Eugene, USA). They were then observed under a microscope that was equipped with a confocal laser-scanning system (LSM 780: Carl Zeiss, Munich, Germany). Alexa488 fluorescence was observed with excitation at 488 nm (Ar laser) and emission at 495 to 540 (Figure S1). Lignin autofluorescence was not detected at these emission wavelengths. The instrument settings were unchanged when comparing fluorescence levels between samples.

For alkaline phosphatase staining [16], leaves were delignified in 15 mL of 8% sodium chlorite solution containing 1.5% acetic acid by shaking at 50 rpm at 40 °C for 40 h. The leaves were then washed with water, extracted with 0.1 M KOH containing 0.1% NaBH$_4$ by shaking at 50 rpm at 40 °C for 24 h, and washed again with water. After blocking for 1 h with 3% (*w/v*) nonfat dry skim milk in sodium phosphate-buffered saline (PBS), the leaves were incubated with a 10-fold dilution of monoclonal antibody CCRC-M1 (CarboSource) for 2 h, washed with PBS containing 0.05% Triton X-100, and then

incubated with a 50-fold dilution of anti-mouse IgG conjugated with alkaline phosphatase for 1 h in darkness. The samples were then washed with PBS containing 0.05% Triton X-100 and stained with the alkaline phosphatase staining kit (Sigma-Aldrich, Tokyo, Japan).

## 3. Results

### 3.1. Movement of Radiocesium in Poplar

Here, we show that radiocesium can enter directly into the surface of plant body through the leaves and bark. When 1 µL of 0.1 µM [$^{137}$Cs]CsCl (2.5 MBq/mmol) solution was placed onto the leaf surface of a wild-type poplar seedling, the radionuclide spread throughout the entire leaf in 10 min, moving to the next (upper) leaf in 30 min, and then to the next upper leaves in 60 min. It then spread throughout the leaves, stems, and roots within 24 h (Figure 1A). When radiocesium was placed on the surface of the stem, it was incorporated within 10 min, moving to the nearest leaves and higher stems in 30 min, and then to the higher stems and the bases of the topmost leaves in 60 min. It then spread throughout the stems, leaves, and roots within 24 h.

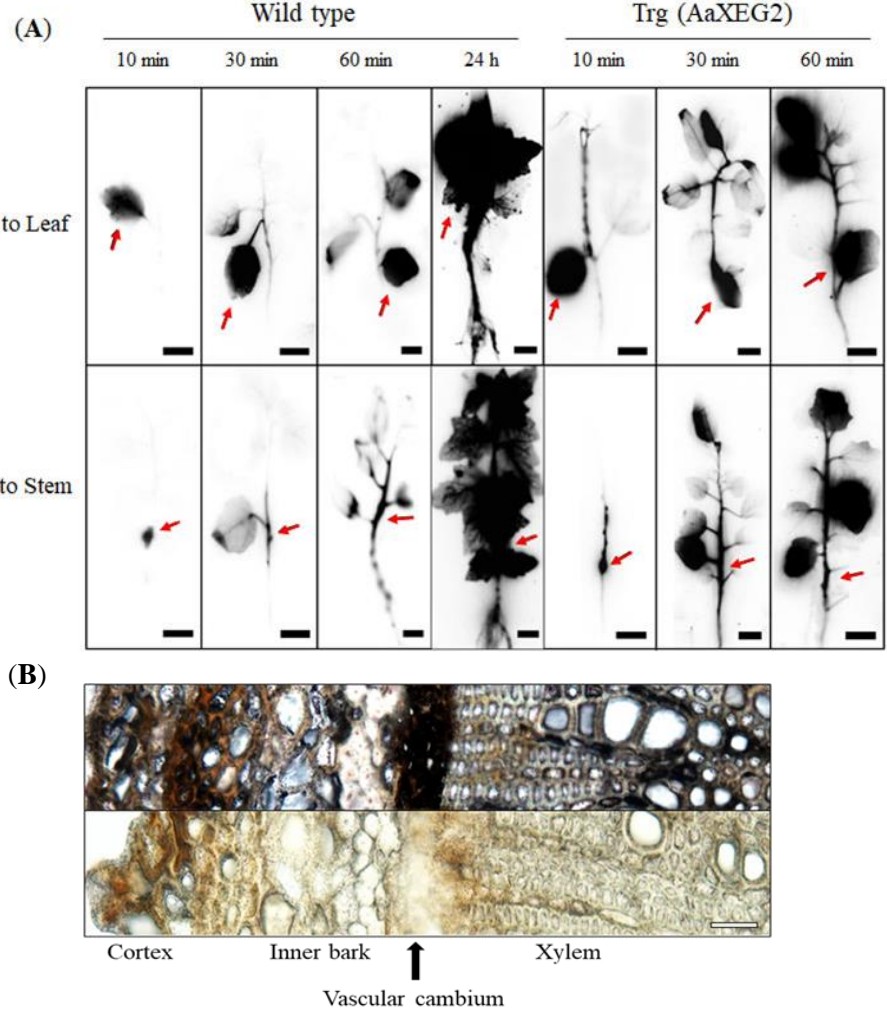

**Figure 1.** Autoradiographs of white poplar. (**A**) Time-course series of autoradiographs showing $^{137}$Cs movement in poplar seedlings after the application of a [$^{137}$Cs]CsCl solution (1 µL) either to the leaf or bark. The red arrows indicate the application points. Two types of poplars were analyzed in the experiments: The wild-type and the transgenic poplar overexpressing xyloglucanase [Trg (AaXEG2)]. Bars = 2 cm. (**B**) Transverse section of a poplar stem after the application of a [$^{137}$Cs]CsCl solution (upper) and a control section (lower). Bar = 100 µm.

The question is whether radiocesium passes through pit membranes in the body. Using transgenic poplars overexpressing xyloglucanase, radiocesium that was attached to the leaf moved faster to the top of the stem without leaves compared with wild-type poplars, and radiocesium attached to the stem moved to the longer upper stem in 10 min (Figure 1A). The movement of radiocesium was faster in the longitudinal direction of the transgenic poplar stems compared with the wild-type stems. In the transgenic poplars, xyloglucan was specifically degraded, as shown by a decrease in 4,6-linked glucosyl linkages (Table S1). Longitudinal sections of their xylem were not stained with an antibody CCRC-M1 against xyloglucan [15], in which xyloglucan appeared mostly in the pit membranes and young leaf wall layer in the wild-type poplars (Figure S1).

Autoradiography of the transverse stem section of wild-type poplar showed that dark silver grains indicating $^{137}$Cs were distributed within 30 min in all the tissues, from the surface, cortex, and inner bark to the vascular cambium (Figure 1B). The silver gains were also observed in ray cells and in parenchyma cells around vessels in the xylem. This distribution pattern of grains indicates that $^{137}$Cs moves through the symplastic pathway from the epidermis to the xylem, where the radionuclide markedly accumulates in the vascular cambium. In the xylem, the silver grains were localized in the parenchyma cells around vessels and were also able to spread in the longitudinal direction. This agrees with the observation (Figure 1A) that $^{137}$Cs deposited on the epidermis moved to the upper stem within 30 min.

### 3.2. Examination of Perforated Xylem as Artificial Heartwood

Based on the radiocesium accumulation in the forest trees, the levels of radiocesium were relatively increased in the heartwood and roots of trees. Using poplar seedlings, higher levels of radiocesium were localized in the dead cells caused by such perforation and $H_2O_2$ sterilization, which represented artificial changes in poplar (Figure 2A).

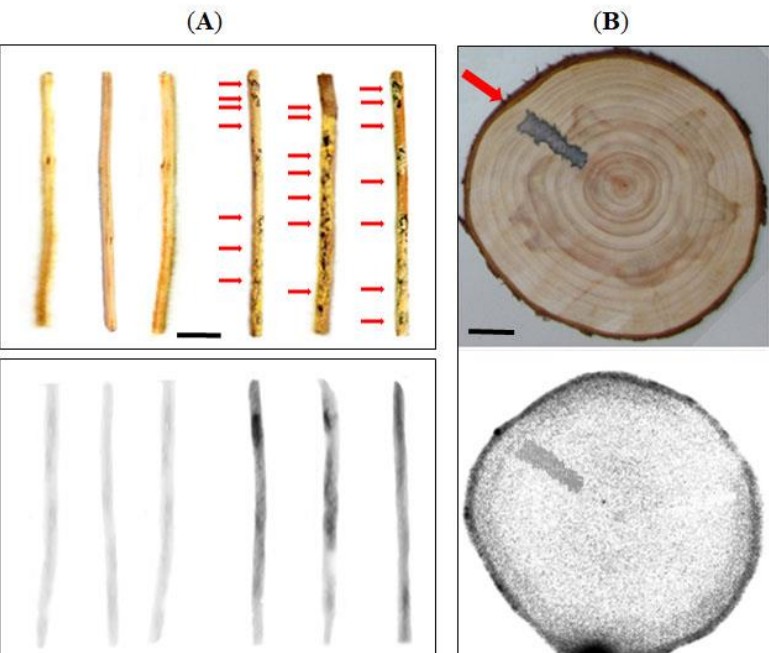

**Figure 2.** Perforated xylem and autoradiographs. Red arrows show perforated parts. (**A**) The xylem of white poplar seedlings for three control xylem and three perforated xylem (upper) and their autoradiographs (lower). Bar = 1 cm. These seedlings were cultured for one week after the application of 1 μL of 0.1 μM [$^{137}$Cs]CsCl (2.5 MBq/mmol) to the surfaces of their main stems. Then, they were cultured under watering conditions with 20 mM KCl solution for one additional month. (**B**) Transverse section of cypress stem with its autoradiograph in the forest field. The stem was diagonally perforated and cut down after 10 months. Bar = 2 cm.

The radiocesium was also localized in the dead cells, which represented the artificially changed sapwood of cypress in Fukushima forest (Figure 2B). This is in agreement with the observation [5] that the movement of radiocesium could accumulate in the heartwood of the tree stem. The radionuclide could also be further deposited in trees containing high levels of potassium. Nevertheless, this cypress wood seemed susceptible to the radionuclide accumulation through the damaged sapwood even at low potassium content (0.98 mg per kg wood) in the outer ring. The autoradiographic image-plate pattern of transversely cut section evidently showed artificial heartwood in the forest trees. Under this condition, no signals were observed from the stored trunks that were cut before the accident.

### 3.3. Flow of Radiocesium

The seedlings used here were diagonally perforated at seven places in the stems with a drill, followed by $H_2O_2$ sterilization (Figure 2A) [8]. Here, we show the flow of radiocesium in the whole body of poplar seedlings containing the places as an artificial heartwood. We also propose that young seedlings with a perforated stem could be useful for the forest tree model.

The perforated seedling main stem surfaces were treated with 1 µL of 0.1 µM [$^{137}$Cs]CsCl (2.5 MBq/mmol) in 20 mM MES/NaOH buffer (pH 6.2) approximately 2 cm above the soil, and then the seedlings were cultured for one week (control condition). The seedlings were then further cultured in pots for one additional month under one of four conditions: (1) low temperatures less than 8 °C along with a limited duration of light (8 h per day) (Table S3); (2) watering with 20 mM KCl solution (Table S4); or (3) new leaf formation after removal of foliage in the seedlings that were watered with 20 mM KCl solution (Table S5) or (4) new leaf formation after removal of foliage in control condition (Table S6).

Significant movement of radiocesium was observed from the vascular cambium and apical stem to the leaves and xylem in a poplar seedling, either at low temperatures (less than 8 °C) along with a limited duration of light or by potassium fertilization in the soil (Figure 3 and Tables S3 and S4). In the case of the potassium fertilization, the movement was increased into the perforated xylem from $2.52 \times 10^3$ cpm to $9.66 \times 10^3$ cpm. The xylem left a certain amount of radiocesium, even if a large amount of the radionuclide moved into the new leaves that grew after removal of the foliage (Figure 3 and Tables S5 and S6). We have confirmed that no turnover occurred in radiocesium after it enters the artificial heartwood.

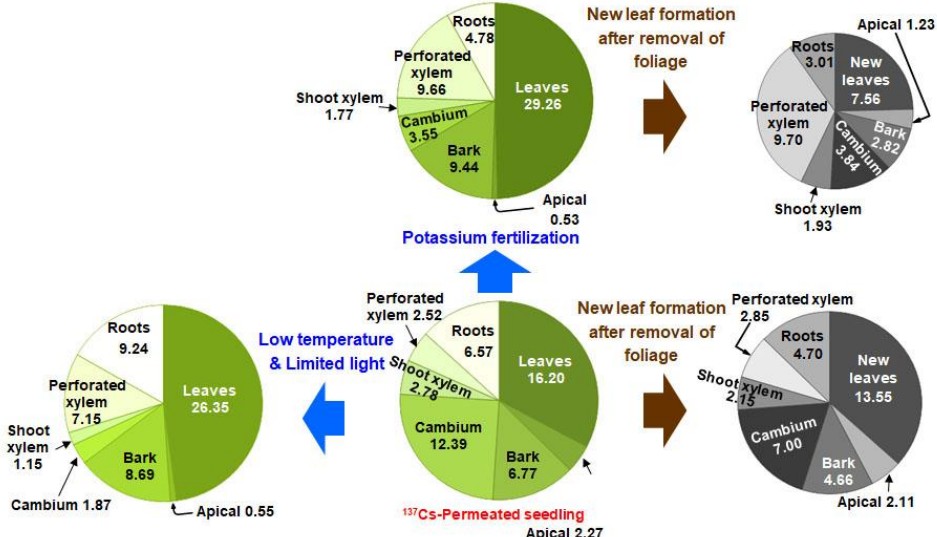

**Figure 3.** Changes in the amount of radiocesium in tissues of differing poplar seedlings. Amounts of radiocesium were shown as radioactivity ($\times 10^3$ cpm) in tissues, resulting from perforated stem seedlings (Tables S2–S6). Cambium, vascular cambium; perforated xylem, perforated stump xylem. The size of pie chart shows the content of radiocesium in each poplar.

## 4. Discussion

Here, we clearly showed the direct incorporation of radiocesium into white poplars using [$^{137}$Cs]CsCl solution in the lab (Figures 1–3). Similarly, fallout radiocesium has potentially been incorporated as an aerosol into the surface of the forest trees in rural areas. The more radiocesium that was deposited in the area of Fukushima, the more the forest trees absorbed it and the more radionuclide can move to the meristic growth zones, as shown in Figure 1. Therefore, growing meristematic cells can incorporate the radionuclide into the trees. For trees that incorporated extremely high radiocesium levels, some might stop growing, as shown in the case of Chernobyl [17], or bend their stems, as found in the atomic-bombed trees in Hiroshima [18], which resulted from high radiation levels.

Radiocesium can move to the growing cells around meristems to infiltrate the seedlings from the outer bark to the inner bark. Radiocesium can further migrate to the xylem from the sapwood to the heartwood through the symplastic pathway and can also migrate in the longitudinal direction of the stem through the apoplastic pathway (Figure 1). After cell growth has ceased, high levels of radiocesium turnover have not been observed in the xylem. Thus, a heartwood might be the final migration place for the radionuclide.

The radiocesium in the meristems of stems passes mainly into the abscission tissues such as leaves and heartwood during induced potassium incorporation and reduced seasonal growth. It is unclear whether the cambium cells and the ray cells of sapwood use the same potassium transporter for transportation in and out of the cells [19]. The dead cells in the xylem could be used as refuse depositories for saturated and waste metabolites, although Stewart [20] suggested that cell death results from the accumulation of the metabolites during heartwood formation. Therefore, heartwood might be the end destination for radiocesium as an excretory metal ion. The sapwood ray cells could serve as translocation channels from the cambium to the heartwood.

## 5. Conclusions

Radiocesium can enter directly into leaves and bark and then accumulates in the cambium. During seasonal growth, the radionuclide passes into abscission tissues such as leaves and heartwood. The heartwood is the final migration place of forest trees for radiocesium.

**Supplementary Materials:** The following are available online at http://www.mdpi.com/1999-4907/10/9/736/s1. Table S1: Composition and methylation analyses of tissues obtained from transgenic poplars (Trg) overexpressing xyloglucanase (AaXEG2), Table S2: Radiocesium distributions in poplar seedling, Table S3: Radiocesium distributions in poplar seedlings after exposed to low temperature along with a limited duration of light, Table S4: Radiocesium distributions in poplar seedlings after potassium fertilization, Table S5: Radiocesium distributions in poplar seedlings on new leaf formation after potassium fertilization and removal of foliage, Table S6: Radiocesium distributions in poplar seedlings on new leaf formation after defoliation, Figure S1: Immuno-staining images of longitudinal section of poplar xylem (left) together with a bright-field image (middle) and whole leaf (right) with the anti-xyloglucan antibody CCRC-M1.

**Author Contributions:** Conceptualization and methodology, T.H.; investigation, S.A., M.N., C.Y., M.I., M.T., K.B., H.O., T.S., I.U., R.K., T.T., Y.S., T.H.; writing, T.H.; funding acquisition, T.T., Y.S., and T.H.

**Funding:** This work was supported by a grant from Tokyo University of Agriculture for the Eastern Japan Reconstruction Support Project after the Fukushima Disaster.

**Acknowledgments:** We are grateful to Tatsuhiko Kodama for the use of $^{137}$Cs in the Radioisotope Center at the University of Tokyo and to the foresters in Soma and Minamisoma.

**Conflicts of Interest:** The authors declare no conflict of interest.

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
