# Peer review of "Intake of Radionuclides in the Trees of Fukushima Forests 2. Study of Radiocesium Flow to Poplar Seedlings as a Model Tree"

_forests, doi:10.3390/f10090736_

Round 1

Reviewer 1 Report

The manuscript entitled « Intake if radionuclides in the trees of Fukushima forests. 2. Radiocesium flow” aims at exploring the movement of cesium inside white poplar seedlings at the organ and tissue levels. Although this manuscript is raising the interesting and important question of the future of radionuclides in trees wood, I have several points that should be improved before publication.

1. I found that the manuscript title is misleading. Indeed after reading it I expected a field study and not a lab study (for most of it). The authors should change the manuscript’s title to support their findings.

2. The authors should justify better in the manuscript why they used perforated xylem instead of natural xylem. I didn’t understood the advantages of using such an invasive technique.

3. Please revise the lettering in Figure 1.

4. A more detailed description if the figure 3 is needed. This figure is full of information but there is almost any description of it. Moreover, in the figure 3 the authors are referring to “removal of foliage” on the Figure 3 but there is no information about this point in the Material and Methods’ part (which leaves were removed, how many?...).

5. The authors should also indicate the % instead of the amount of radiocesium on their different pie charts. This will make the figure clearer.

6. The authors choose to use a xyloglucan-deficient line in the first part of their study but why not also using this line for the rest of the experiments and especially the radiocesium flow experiment? Since they are showing differences in the time-course experiment, it will be interesting to follow the radiocesium flow in this line too and to discuss it in an appropriate manner. At least, the authors should discuss the phenotype observed in the AaXEG2 line regarding the increased radiocesium transport across the different organs.

Author Response

Thank you for your review.

change the manuscript’s title: As suggested, we changed the second title of the manuscript, from “2. Radiocesium flow” to “2. Study of radiocesium flow into poplar seedlings as a model tree”.

why they used perforated xylem instead of natural xylem: We added a new sentence, as shown below, in section 3.3. Flow of Radiocesium.

“The seedlings used here were diagonally perforated at seven places in the stems with a drill, followed by H2O2 sterilization (Figure 2A) [8]. Here, we show the flow of radiocesium in the whole body of poplar seedlings containing the places as an artificial heartwood. We also propose that young seedlings with a perforated stem could be useful for the forest tree model.”

Lettering in Figure 1: As suggested, we revised the lettering in Figure 1 to improve the clarity.

A more detailed description if the figure 3 is needed: As suggested, we completely revised the description in section 3.3. Flow of Radiocesium. Please see the revised manuscript.

5.also indicate the % instead of the amount of radiocesium on their different pie charts: We believe that the radioactivity due to the pie chart is relatively easy to see because total amounts of radiocesium in each poplar are shown as the size of pie chart. We added "The size of pie chart shows the content of radiocesium in each poplar" in the legend of Figure 3.

xyloglucan-deficient line: The reason we used the xyloglucan-deficient poplar (transgenic poplar overexpressing xyloglucanase) was because xyloglucan could be the main component of pit membranes, and radiocesium can travel through these membranes. The results showed the involvement of pit membranes in the flow of radiocesium. However, we did not use the transgenic line that was used in other experiments. The transgenic poplar phenotype is shown in reference 7.

Reviewer 2 Report

Reviewer comments for the manuscript entitled, Intake of radionuclides in the trees of Fukushima forests

General comments

The manuscript is quite written and easy to understand. It covers an important subject in the area and it is well covered.

Detailed comments

Title

Check the title, sounds not total correct. Why a “2” in line 4? Is the manuscript part of a article series?

Abstract

Line 19: “a” is missing, “a large area …”

Line 20: cesium-137, in the manuscript you are using sometimes 137Cs, cesium-137 and radiocesium-137. Please be consequent and use the same form all the time.

Keywords

No comments

Introduction

Overall comments. The introduction is to short and it starts with your own results (Lines 32 – 35). It had been good to expand the introduction and explain more why you are doing this study and the importance of it.

Lines 36-37: Missing reference.

Line 38: Remove the word “intensive”

Line 39: Was it really extremely high levels of radiation in the study? 2.5 MBq cesium is quite high but not extremely.

Line 41: Rephrase

Line 42-44: Rephrase

Line 45-46: Missing reference

Materials and Methods

This section has good details but are sometimes hard to follow.

Line 50: The scientific name of poplar is missing.

Line 53: Add year when obtained the CsCl

Line 54 – 55: How did you placed the buffer?

Line 57: Did you used tap water or deionized water?

Line 57: What type of microwave did you used? Level?

Line 58: Add Tokyo, Japan to the Fujifilm

Lines 61 – 67: This section is a bit confusing. Can you e.g. write like this: 1) Low temperatures…, 2) Watering with…, 3) New leaf… And so on.

Line 67-68: How many (in numbers) is “several cultivations”?

Line 70: I don’t understand the last part of this sentence.

Line 85: The scientific name is missing for cypress and what species of cypress.

Line 89-90: Maybe a picture is good to show?

Line 98: Cultured for 30 min? I think better not use the word “cultured” when it is only 30 min.

Line 100: The water again. What kind of water?

Line 103: Now you are talking about Japanese cedar. This three was not mention before. Maybe mixed up? I think it should be cypress.

Line 107: What was developed?

Line 114: Maybe add “cell”

Line 117: What kind of water?

Results

The results from this study are very interesting and this section is clear and understandable.

Line 195: 137Cs should be 137Cs

Line 224: Decide how you are writing cesium-137.

Discussion

Line42: “In like fashion” sounds strange, maybe rephrase.

Line 248: “strong” is not so specific maybe not use that word.

Author Contribution

This section is missing

Funding

This section is missing

Acknowledgments

No comments

References

No comments

Figure legends

No comments

Supplemental Tables 1 to 5

No comments

Author Response

Thank you for your review.

Title: Why a “2” in line 4? Is the manuscript part of a article series?: Yes, this manuscript is part of a series; the first manuscript in the seriesis “1. Field study”. We changed the second title of the manuscript to “2. Study of radiocesium flow into poplar seedlings as a model tree”.

Abstract: Line 19: As suggested, we added “a”.

Abstract: Line 20: We have changed "cesium-137" to "radiocesium".

Introduction: explain more why you are doing this study: As suggested, we provided more explanation of our study in the Introduction.

Lines 36-37: Missing reference: We changed the reference number from 3 to 5. Thus, the proper references are now included.

Line 38: Remove the word “intensive”: As suggested, we deleted “intensive”.

Line 39: Was it really extremely high levels of radiation in the study? 2.5 MBq cesium is quite high but not extremely: As suggested, we changed “extremely high radiation” to “radiocesium”.

Line 41: Rephrase: We were not able to determine the text that required rephrasing.

Line 42-44: Rephrase: We were not able to determine the text that required rephrasing.

Line 45-46: Missing reference:As suggested, we added the reference.

Materials and Methods

Line 50: The scientific name of poplar is missing: As suggested, we added “Populus alba L.”.  

Line 53: Add year when obtained the CsCl: As suggested, we added the year “in 2013”.

Line 54 – 55: How did you placed the buffer?: We dissolved [137Cs]CsCl in the buffer.

Line 57: Did you used tap water or deionized water?: We always used either deionized water or Millipore-filtrated water in the experiments.

Line 57: What type of microwave did you used? Level?: As suggested, we added “Panasonic NE-KM264-FG1000 W”.

Line 58: Add Tokyo, Japan to the Fujifilm: As suggested, we added “Tokyo, Japan”.

Lines 61-67: As suggested, we used the suggested text as follows:“1) low temperatures…; 2) watering with…; or 3) new leaf…; 4) 4) new leaf formation...”.

Line 67-68: How many (in numbers) is “several cultivations”: We always used three seedlings for each experiment.

Line 70: I don’t understand the last part of this sentence: We changed the sentence from “the cambium was collected by scraping off both the surface of the xylem and the inside of the bark with a blade from the shoots and basal perforated stem” to “the cambium was collected by scraping off both the surface of the xylem and the inside of the bark with a blade from the stem”.

Line 85: The scientific name is missing for cypress: As suggested, we added “Chamaecyparis obtusa”.

Line 89-90: Maybe a picture is good to show: We do not agree that a picture would be included in the manuscript.

Line 98: Cultured for 30 min? I think better not use the word “cultured” when it is only 30 min: As suggested, we deleted the sentence, “After 30 min by the application of…”.

Line 100: The water again. What kind of water?: We used deionized water here. Because it is commonly known that either deionized water or Millipore-filtrated water would be used in the experiments.

Line 103: Now you are talking about Japanese cedar. This three was not mention before. Maybe mixed up? I think it should be cypress: We apologize for the error. We deleted the section, “and a microslicer for Japanese cedar”. Also, we deleted the sentence, “and control Japanese cedar stems were obtained from Mie Prefecture, 500 km distance from Fukushima”.

Line 107: What was developed?: As suggested, we changed “They were” to “The films were”.

Line 114: Maybe add “cell”: As suggested, we added “Cell”.

Line 117: What kind of water?: We used deionized water here.

Results

Line 195: 137Cs should be 137Cs: As suggested, we changed to “137Cs”.

Line 224: Decide how you are writing cesium-137: As suggested, we have not used “radiocesium-137”.

Discussion

Line 242: “In like fashion” sounds strange, maybe rephrase: As suggested, we changed to “Similarly”.  

Line 248: “strong” is not so specific maybe not use that word: As suggested, we changed the wording to “high levels of”.

Author Contribution and Funding

This section is missing: We added “Author Contribution” and “Funding” sections.

Reviewer 3 Report

Dear Editor,

As I understood while reading, the manuscript entitled "Intake of radionuclides in the trees of Fukushima forests 2. Radiocesium flow". (forests-524567) appears to be the companion manuscript of another one.

The manuscript starts from field observations on forest trees, reported in the companion manuscript, to investigate radiocesium movement and the influence of K ions on uptake during experimental uptake in poplar seedlings.

The Authors did a lot of work but experimental plans are not clearly explained and are mostly reported in the Materials and Methods section. It is therefore very hard to follow the text flow and to understand results: great effort is needed to guess which experiment figures are reporting. Interpretation of the results is often missing and Conclusion does not summarize all result and their interpretation. Nevertheless results are interesting and sound, and merit to be published without additional work. The subject is also of general relevance not only to the scientific community.  For these reasons I did all my best to help the Authors understand all the changes needed to present their results in a better and clearer way.  

As said, the main flaw of the manuscript is that the rationale and design of the experiments is not reported at the beginning of each experimental section but rather described in the Methods sections, which should just describe technical procedures. The reader must deduce the experimental procedure from the Material and methods section and this is quite confusing. Mistakes and misprints are present here and there. It is likely that the extant manuscript has been repackaged from a previous version without great care. Some flaws in English usage also impair understanding.

The manuscript could be acceptable only after careful and extensive revision. I tried to signal all the many things that need to be fixed before the article could be accepted, but the Authors must do their part to guarantee an acceptable final result. In particular, pay attention that each experimental results refer to a Figure, Table or to not shown data.

In detail:

Introduction

Most of the introduction refers to data from the companion paper concerning the field study but this is not clearly indicated. Very few works on the subject are mentioned. Is really the background so limited?

Lines 32-35:  please cite a reference. I believe it refers to the companion paper but it is not clear at all.

Lines 36-37:  Growing….field. the sentence is not clear, please rephrase

Line 38: It is not clear to me if “This paper” refers to the submitted manuscript or to the companion manuscript related to the Field study. Please clarify

Line 36: the word “assembly” should be better replaced by “accumulation”

Lines 37-38: Do this sentence refer to the work reported in the first manuscript or to a general lack of information about radiocesium infiltration in trees? If so, I found some detailed information in the following reference: Masumori et al, “Radiocesium in timber of Japanese cedar and Japanese red pine in the forest of Minamasoma, Fukushima”. In Agricultural implications of the Fukushima nuclear accident., Nakanishi and Tanoi, 2016. I think it should be taken into account also in the Discussion.

Materials and Methods

§ 2.2: please add information about seedling age or height, and the way they were germinated, if different from that reported at 2.4

§ 2.4 and 2.3 should be inverted

Lines 60-67:  This section is mostly related to the experimental design; it would be convenient to move this part in the result section (§ 3.2). Since the experiment is quite complex: different treatments, times etc, a schematic drawing would be of help.

Line 66: Please indicate the control conditions: temperature, light/dark cycle…

Lines 67-68: what does it means “after several cultivations”? Would you mean after the different treatments? The list of the component tissues must be given somewhere (in the result section)

Lines 74, 83 and others: use “untreated” or “control” rather than “natural”

Line 86: References to a Figure of another paper is feasible only if the two manuscript are published together

Lines 91-92 and 95-96: These sentences refer to a result, not a method and should be moved away

Lines 103, 106-107 and 111-112: which Japanese cedar?  No mention at all in the result section!

Lines 112-113: this is a repetition of lines 106-107

§ 2.6 I really do not understand the relevance of this analysis in the context of the manuscript (see later comment of the result section). My suggestion is to limit the description of the method and related result in Table 1 to the components directly related to xyloglucan component. In alternative Table 1 could be shown as a Supplementary Table.

Line 114, 117 and others: Please write “cell wall” rather than just “wall”

Line 142: Figure 1B, not 2B

Results

In general, the titles of the experimental sections should explain the scope of the experiment: “Movement of radiocesium in trees” is acceptable, “Perforated xylem” doesn’t mean anything.

§ 3.1: To avoid confusion, it would be more clear if the authors always refer to poplar as “seedlings” and use the term “tree” for the cypress from the forest

Please start the paragraph with a couple of lines explaining the scope of the experiment and why the transgenic poplar line has been used. While reporting results it would be useful to specifically refer to the different sections of Fig 1A (upper or lower, left or right)

Figure 1:  The title is misleading: just 2 are autoradiographs, while Fig 1 B refers to immunostaining. There is also a mistake inj the legend: white and not which poplar.

Why is the Figure split in two different pages, separated by test and a Table? Maybe this was just a formatting problem. Moreover, if I correctly understand, FIG 1A and B refer to the same experiment or at least to the same plant material (wt and transgenic poplar seedlings) while FIG 1 C refers to a different Cs treatment on different plant material (that reported in M&M paragraph 2.5). This must be clearly stated in the text.

In Figure 1B it is not clear what the central and right pictures refer to? Is this the alkaline phosphatase staining described in M&M? If so, it isn’t written anywhere.. If so, please explain the need of two different detection method for the primary antibody. If not, please explain. No reference to the leaf is present in the text.

Line 172: Please specify that this observation is valid for both leaf-treated and stem-treated seedlings.

line 174: is this citation correct concerning the use of antibody?

Line 175: the interpretation of the different rate of Cs movement in w.t. and transgenic plants, although may seem obvious to Authors, must be given here.

Table 1: I think this Table redundant, since it only confirms the reduction of xyloglucans in transgenic plants, that is already clear from fig 1B. All the other cell wall components analysed do not seem to be of any relevance for the manuscript. The table could be eventually shown as Supplementary material.

Line 197 and Fig 1C: Which poplar stem sections? W.t, control, treated, perforated?  The reader is led to think that Authors are speaking of the same experiment of Fig 1A and 1B but it does not seem so. Cesium treatment seems to be that described in material and methods section 2.5 but this is not specified in the text. Again, the experiment and its scope should be briefly described at the beginning of each result paragraph.

§ 3.2: Again, experiment and its scope are not reported: why and what have been done must be written in the result section before reporting results, while how was it done is a matter of M&M section. Why have stems been perforated? What is the aim of this experiment? Any reader, even if not expert in the field, should be able to understand the experimental logic. The experimental design reported in m&M should be moved here. As already suggested, a schematic drawing could be of help.

Lines 208-210: what does this sentence refer to? As far as I understand, radiocesium was applied to the outer stem, not to leaves or roots in this experiment, supposing the Authors refer to the treatment described in the first three lines of § 2.3.

Lines 213-214: Where this data about K content come from? If it is a not shown data, this need to be specified.

Figure 2B: to which of the several sections of the tree described in§ 2.4 do the figure refers to?

Line 228: “…perforated at ten places….” Both in Fig 1A and at line 79, seven holes are shown/described

Lines 229-230: Where is it shown that the radiocesium movement was increased due to potassium fertilization in perforated xylem but not in the control xylem?

Lines 231-232: The reference to Figure 3 Is missing

Figure3: The sole mention to this important Figure refers to Cesium transport to leaves, but other relevant differences in Cesium distribution upon the different treatments can be seen in the pie charts and should be commented.

Discussion: This section should summarize into more detail the results and their interpretation.

I’m not sure that this long list is exhaustive, so please read carefully all the text to correct typing errors, wrong tenses or misuse of some words and other minor mistakes.

Author Response

Thank you for your review.

Introduction

Most of the introduction refers to data from the companion paper concerning the field study but this is not clearly indicated. Very few works on the subject are mentioned. Is really the background so limited?: We added more description into the Introduction (please see text).

Line 32-35: As suggested, we revised the references to improve the clarity.

Lines 36-37: Growing….field. the sentence is not clear, please rephrase: We changed the sentence (please see the text).

Line 38: It is not clear to me if “This paper” refers to the submitted manuscript or to the companion manuscript related to the Field study. Please clarify: “This paper” means this manuscript not the accompanying paper. I think that this term will be understandable for the readers. If not, please provide an explanation.

Line 36: the word “assembly” should be better replaced by “accumulation”: As suggested, we changed “assembly” to “accumulation”. Thank you for nice comment and the clear suggestion.

Lines 37-38: Do this sentence refer to the work reported in the first manuscript or to a general lack of information about radiocesium infiltration in trees? If so, I found some detailed information in the following reference: As suggested, we added the reference (below). Many thanks for your comment and definite suggestion.

6) Masumori, M.; Nogawa, N.; Sugiura, S.; Tange, T. Radiocesium in timber of Japanese cedar and Japanese red pine, in the forests of Minamisoma, Fukushima. In Agricultural implications of the Fukushima nuclear accident. Nakanishi, T.M. & Tanoi, K., eds (Tokyo, Japan: Springer), pp. 161-174, 2016.

Materials and Methods

2.2: please add information about seedling age or height, and the way they were germinated, if different from that reported at 2.4: As suggested, we added the information about poplar seedlings (Please see the text).

2.4 and 2.3 should be inverted: As suggested, we switched the numbers 2.4 and 2.3.

Lines 60-67:  This section is mostly related to the experimental design; it would be convenient to move this part in the result section (§ 3.2). Since the experiment is quite complex: different treatments, times etc, a schematic drawing would be of help: As suggested we moved this section to “3. Results”.

Line 66: Please indicate the control conditions: temperature, light/dark cycle: The control condition is the cultivation condition for poplar seedlings.

Lines 67-68: what does it means “after several cultivations”? Would you mean after the different treatments? The list of the component tissues must be given somewhere (in the result section): “Several cultivations” means five cultivations, as shown in Figure 3.

Lines 74, 83 and others: use “untreated” or “control” rather than “natural”: As suggested, we used “control” alone.

Line 86: References to a Figure of another paper is feasible only if the two manuscript are published together: As suggested, we employed the reference number alone.

Lines 91-92 and 95-96: These sentences refer to a result, not a method and should be moved away: We moved the sentences to section “3. Results”.

Lines 103, 106-107 and 111-112: which Japanese cedar?  No mention at all in the result section!: We apologize for the error. We deleted the section, “and a micro slicer for Japanese cedar”.

Lines 112-113: this is a repetition of lines 106-107: We also deleted the sentence, “and control Japanese cedar stems were obtained from Mie Prefecture, 500 km distance from Fukushima”.

2.6 I really do not understand the relevance of this analysis in the context of the manuscript (see later comment of the result section). My suggestion is to limit the description of the method and related result in Table 1 to the components directly related to xyloglucan component. In alternative Table 1 could be shown as a Supplementary Table: As suggested, Table 1 was removed from the main part of the manuscript and included as Supplementary tables.

Line 114, 117 and others: Please write “cell wall” rather than just “wall”: As suggested, we changed “wall” to “cell wall”.

Line 142: Figure 1B, not 2B: As suggested, we changed the figure to “Supplemental Figure 1”.

Results

In general, “Perforated xylem” doesn’t mean anything: As suggested, we changed “Perforated xylem” to “Examination of Perforated Xylem as Artificial Heartwood”.

3.1: To avoid confusion, it would be more clear if the authors always refer to poplar as “seedlings” and use the term “tree” for the cypress from the forest: As suggested, we used “seedlings” for poplar and “tree” for cypress in the text.

Please start the paragraph with a couple of lines explaining the scope of the experiment and why the transgenic poplar line has been used. While reporting results it would be useful to specifically refer to the different sections of Fig 1A (upper or lower, left or right): As suggested, we divided the explanation for Figure 1A into different sections.

Figure 1: The title is misleading: just 2 are autoradiographs, while Fig 1 B refers to immunostaining. There is also a mistake inj the legend: white and not which poplar: We corrected the error in the legend. We also moved Figure 1B (immune-staining) to the Supplementary figures.

Why is the Figure split in two different pages, separated by test and a Table? Maybe this was just a formatting problem. Moreover, if I correctly understand, FIG 1A and B refer to the same experiment or at least to the same plant material (wt and transgenic poplar seedlings) while FIG 1 C refers to a different Cs treatment on different plant material (that reported in M&M paragraph 2.5). This must be clearly stated in the text: We revised the manuscript completely (please see the text).

In Figure 1B it is not clear what the central and right pictures refer to? Is this the alkaline phosphatase staining described in M&M? If so, it isn’t written anywhere. If so, please explain the need of two different detection method for the primary antibody. If not, please explain. No reference to the leaf is present in the text: Figure 1B was moved to the Supplemental figure, where the two different staining procedures can be shown.

Line 172: Please specify that this observation is valid for both leaf-treated and stem-treated seedlings: We changed the sentence, as follows:“The question is whether radiocesium passes through pit membranes in the body. Using transgenic poplars, radiocesium that was attached to the leaf moved faster to the top of the stem without leaves compared with wild type poplars and radiocesium attached to the stem moved to the longer upper stem in 10 min (Figure 1A). The movement of radiocesium was faster in the longitudinal direction of the transgenic poplar stems compared with the wild-type stems."

Line 174: is this citation correct concerning the use of antibody?: We corrected the reference number.

Line 175: the interpretation of the different rate of Cs movement in w.t. and transgenic plants, although may seem obvious to Authors, must be given here: As shown above, we changed the sentence.

Table 1: I think this Table redundant, since it only confirms the reduction of xyloglucans in transgenic plants, that is already clear from fig 1B. All the other cell wall components analysed do not seem to be of any relevance for the manuscript. The table could be eventually shown as Supplementary material: As suggested, we moved Table 1 to the Supplementary tables.

Line 197 and Fig 1C: Which poplar stem sections? W.t, control, treated, perforated?  The reader is led to think that Authors are speaking of the same experiment of Fig 1A and 1B but it does not seem so. Cesium treatment seems to be that described in material and methods section 2.5 but this is not specified in the text. Again, the experiment and its scope should be briefly described at the beginning of each result paragraph: As suggested, we revised the explanation of Figure 1B (old Figure 1C) (please see the revised manuscript).

3.2: Again, experiment and its scope are not reported: why and what have been done must be written in the result section before reporting results, while how was it done is a matter of M&M section. Why have stems been perforated? What is the aim of this experiment? Any reader, even if not expert in the field, should be able to understand the experimental logic. The experimental design reported in m&M should be moved here. As already suggested, a schematic drawing could be of help: We added additional text into the introduction of section 3.2, “Based on the radiocesium accumulation in the forest trees, the levels of radiocesium were relatively increased in the heartwood and roots of trees.....”

Lines 208-210: what does this sentence refer to? As far as I understand, radiocesium was applied to the outer stem, not to leaves or roots in this experiment, supposing the Authors refer to the treatment described in the first three lines of § 2.3: We also added the sentence, “For the forest tree cypress in Fukushima, the autoradiographic image-plate patterns of transversely cut sections evidently showed artificial heartwood in the forest trees.”

Lines 213-214: Where this data about K content come from? If it is a not shown data, this need to be specified: We added the new sentence, “The movement of radiocesium could always accumulate in the heartwood of the tree stem, in which the radionuclide could be further deposited in trees containing high levels of potassium.”

Figure 2B: to which of the several sections of the tree described in § 2.4 do the figure refers to?: This is one of the sections for the cypress stem.

Line 228: “…perforated at ten places….” Both in Fig 1A and at line 79, seven holes are shown/described: We apologize for the error. “Seven holes”iscorrect. We revised the text.

Lines 229-230: Where is it shown that the radiocesium movement was increased due to potassium fertilization in perforated xylem but not in the control xylem?: We revised the sentences, “Significant movement of radiocesium was observed from the vascular cambium and apical stem to the leaves and xylem in a poplar seedling, either at low temperatures (less than 8°C) along with a limited duration of light or by potassium fertilization in the soil (Figure 3 and Supplemental Tables 3 to 4). In the case of the potassium fertilization, the movement was increased into the perforated xylem from 2.52 Bq to 9.66 Bq. The xylem left a certain amount of radiocesium, even if a large amount of the radionuclide moved into the new leaves that grew after removal of the foliage (Figure 3 and Supplemental Tables 5 to 6). We have confirmed that no turnover occurred in radiocesium after it enters the artificial heartwood.”

Lines 231-232: The reference to Figure 3 Is missing: We added the reference “(Figure 3)”.

Figure 3: The sole mention to this important Figure refers to Cesium transport to leaves, but other relevant differences in Cesium distribution upon the different treatments can be seen in the pie charts and should be commented: We added additional information for Figure 3.

Discussion: This section should summarize into more detail the results and their interpretation. I’m not sure that this long list is exhaustive, so please read carefully all the text to correct typing errors, wrong tenses or misuse of some words and other minor mistakes: Thank you for your clear and specific comments.